# A Novel Real-Time Processing Wideband Waveform Generator of Airborne Synthetic Aperture Radar

Dongxu Chen [1,2], Tingcun Wei [1,*], Gaoang Li [1,2], Jie Feng [2,3], Jialong Zeng [2,3], Xudong Yang [2] and Zhongjun Yu [2,3]

1   School of Microelectronics, Northwestern Polytechnical University, Xi'an 710072, China; chendx@aircas.ac.cn (D.C.); liga@aircas.ac.cn (G.L.)
2   Aerospace Information Research Institute, Chinese Academy of Sciences, Beijing 100094, China; fengjie@aircas.ac.cn (J.F.); zengjialong19@mails.ucas.ac.cn (J.Z.); yangxd01@aircas.ac.cn (X.Y.); yuzj@ucas.ac.cn (Z.Y.)
3   School of Electronic, Electrical and Communication Engineering, University of Chinese Academy of Sciences, Beijing 100049, China
*   Correspondence: weitc@nwpu.edu.cn

**Abstract:** This paper investigates a real-time process generator of wideband signals, which calculates waveforms in a field-programmable gate array (FPGA) using the high-level synthesis (HLS) method. To obtain high-resolution and wide-swath images, the generator must produce multiple modes of large time-bandwidth product (TBP) linear frequency modulation (LFM) signals. However, the conventional storage method is unrealistic as it requires huge storage resources to save pre-computed waveforms. Therefore, this paper proposes a novel processing approach that calculates waveforms in real-time based simply on parameters such as the sampling frequency, bandwidth, and time width. Additionally, this paper implements predistortion through the polynomial curve to approximate phase errors of the system. The parallelizing process in the FPGA is necessary to satisfy the high-speed requirement of a digital-to-analog converter (DAC); however, repeatedly multiplexing real-time calculation consumes extensive logic and DSP resources, potentially exceeding FPGA limitations. To address this, this paper proposes a piecewise linear algorithm to conserve resources, which processes the polynomial only once, acquires the difference in two adjacent values through the register and pipeline, and then adds this increment to facilitate parallel computations. The performance of this proposed generator is validated through simulation and implemented in experiments with an X-band airborne SAR system.

**Keywords:** LFM; real-time; polynomial; HLS; piecewise linear

## 1. Introduction

Synthetic aperture radar (SAR) serves as a highly influential and widely used remote sensing system, functioning as a high-resolution microwave imaging radar capable of operating nearly full-time and in almost all weather conditions [1–3]. As a significant approach to aerial remote sensing, airborne SAR is extensively utilized in Earth observation, environmental monitoring, strategic target surveillance, disaster monitoring, ground moving target indication (GMTI), and other applications [4–7]. To obtain high-resolution and wide-swath (HRWS) images, SAR typically transmits a linear frequency modulation (LFM) signal and implements matched filtering in range direction [8,9]. Due to the motion of the SAR platform, the signal also possesses the form of LFM in the azimuth direction [10]. Therefore, the high-quality generation of LFM signals is crucial in airborne SAR systems. In a SAR system, the signal generation circuit is mainly based on a field-programmable gate array (FPGA) and digital-to-analog converter (DAC) [11,12]. Considering the characteristics of flexibility and high performance, the FPGA processes complicated signals simply by using the look-up table (LUT) method, which reads stored calculated waveforms or

phase information from memory [13]. The range direction resolution depends on the signal bandwidth [14]; consequently, this requires a very high sampling frequency of the DAC to generate signals. In addition to acquiring various modes of HRWS images, airborne SAR systems also have additional capabilities, such as GMTI. This requires the signal generator to store a large number of combinations of time-bandwidth product (TBP) LFM signals. However, this approach requires substantial dedicated storage resources, which can be inflexible and impractical with limited hardware resources. Hence, the storage method is not suitable for this requirement.

Although waveforms can be synthesized by pre-storing calculated data in ROM or a dedicated flash chip expediently [15], this method is not fit for multi-mode, large TBP SAR systems that consume extensive storage resources. Another commonly used method is direct digital synthesis (DDS), which generates the LFM signal by accumulating frequency information through initial parameters and using it as an address to read signals from memory [16]. When compared with the direct storage of waveforms, the DDS method demands fewer storage resources but is more complicated [17]. These proposed methods are inflexibly implemented and require significant resources to store pre-computed information for multi-mode SAR systems with varying bandwidths and time widths.

To implement within constrained hardware resources flexibly, a real-time processing LFM signal generator based on high-level synthesis (HLS) is proposed [18]. This approach calculates LFM waveforms in real-time using key parameters such as time width, bandwidth, and sampling frequency. Typically, the register-transfer level (RTL) design of FPGAs is implemented using a hardware description language, such as Verilog or VHDL. However, due to the limitations of the design procedure, realizing complex algorithms is challenging [19]. In contrast, the high-level synthesis (HLS) method is more suitable for implementing complex algorithms. It operates in high-level programming languages (C/C++) and translates these into RTL design. HLS is widely applied in fields such as high-performance computing (HPC), deep learning (DL), and signal processing [20,21]. As a result, LFM signal generation can leverage HLS for real-time calculations. However, to conserve DSP and logic resources, this proposed algorithm sequentially processes data flow, which may not satisfy the speed requirements of the DAC. This necessitates caching the calculated results in a high-speed dual-port RAM and then reading parallelized data to the DAC, which is triggered by the pulse repetition frequency (PRF). Although the serial calculation signal generator with real-time predistortion does not require dedicated storage resources, it does consume substantial logic and DSP resources. Particularly, repeatedly multiplexing this module for parallel computing is impractical. Therefore, this approach requires some time to complete the calculating and caching of the waveform before transmitting it to the DAC.

In a high-resolution SAR system, the LFM generator produces in-phase/quadrature (I/Q) signals of a wide bandwidth to simplify the implementation [22]. After they have been filtered through low-pass filters, the I/Q signals are modulated by a quadrature modulation unit in a radio-frequency (RF) circuit to the carrier frequency. This RF signal is then filtered and amplified by a low-noise amplifier [23]. Subsequently, the antenna radiates this full-power RF signal to free space [24]. Simultaneously, the antenna receives reflected echo waves and demodulates the received waves into I/Q signals in the RF circuit, which are then sampled by the ADC and processed. The waveforms are subsequently transmitted to the digital recorder and the signal processing unit. Consequently, the imbalance and nonlinearity of RF transmitting and receiving channels result in signal phase and amplitude distortion, impacting the peak side-lobe ratio (PSLR) and integral side-lobe ratio (ISLR), leading to degradation of the image quality [25,26].

To compensate for the phase errors of the system, the generator should implement predistortion in advance on the storage methods, which can be compensated during the signal generation in the PC. In contrast, the real-time method should compensate during the calculation of phase. System errors are typically approximated using polynomial curves [27]. However, due to the resource constraints of FPGA, processing high-order polynomials

consumes substantial DSP and logic resources, making it impractical to repeatedly reuse this computational module for the multiple data paths of the DAC. Therefore, the phase of the signal with predistortion usually calculates in real-time serially, which results in a computational rate that cannot meet the DAC's sampling frequency requirement. As a result, waveforms demand to be cached in the RAM and be read in parallel using the DAC clock for every PRF pulse.

As a form of radio detection equipment, the rapid progress of high-sensitivity radar signal interception technologies presents a significant risk to many radars when they are activated, increasing the probability of mission failure. To mitigate the risk of radar signals being detected by interceptors and achieve the low probability of intercept (LPI) characteristic of the joint coding waveform, it is necessary to employ frequency modulation or a phase modulation signal [28,29]. This is particularly crucial for specific RF stealth SAR systems that may require each PRF to generate an LFM signal with different parameters, aiming to keep the interceptors in a state of constant uncertainty. Consequently, the storage method and serial real-time calculation approach are unable to meet the timeliness requirements of these systems. Given the limitations of sequential real-time computation, this paper proposes a parallelized real-time algorithm for the simultaneous computation of multiple data path signals.

The required number of parallel data paths varies for different DAC chips. This proposed method can fulfill these requirements without utilizing extra FPGA resources. Furthermore, to compute parallel phase errors in real-time, an enhanced predistortion algorithm is suggested, which employs a piecewise linear approach [30] to approximate the polynomial curve. This algorithm can decrease the resource usage of the parallel signal with predistortion and can be enhanced through pipelining. It is implemented using HLS. The predistortion effect is comparable to the storage and serial real-time calculation method.

For the generation of a signal in a specific operating mode, both the storage method and serial real-time calculation require a significant amount of processing time to complete before the data can be read by the DAC to generate the waveform. In contrast, parallel real-time calculation can process the data shortly after receiving the operating instructions, and then the DAC clock can read and send the data. In complex scenarios that require signals with different parameters for different PRFs, the storage method is impractical. Serial real-time calculation necessitates multiple reuses of modules and consumes logic resources exceeding the limits of the FPGA. Thus, within the limited logic resources available, parallel real-time calculation is the only method that can meet such requirements.

This paper demonstrates the accuracy of the proposed algorithm through simulations and experiments. The experiments are conducted using an X-Band radar, employing a digital synthesis circuit comprising primarily FPGAs, ADCs, and dual-channel DACs. To validate the signal quality, in the SAR closed-loop test mode, the modulated RF signal is looped back to the RF receive circuits, and the demodulated I/Q signals are sampled through ADC. After FPGA processing, the data are recorded using a digital recorder. The paper is organized as follows: Section 2 primarily addresses common signal generation methods, improved parallel real-time signal processing, experimental parameters, and the experiment platform. The simulation and experimental results are presented in Section 3, along with data analysis and a comparison of results using different methods. The discussion is presented in Section 4, and the conclusions are provided in Section 5.

## 2. Materials and Methods

### 2.1. Introduction of Common LFM Generation Methods

Usually, SAR transmits a large-bandwidth LFM signal in the range direction; after being modulated on carrier frequency $f_0$ by the RF circuit, the signal can be expressed as

$$s(\tau) = rect(\frac{\tau}{T_r})\cos(2 \cdot \pi \cdot f_0 \cdot \tau + \pi \cdot kr \cdot \tau^2), \quad -\frac{T_r}{2} \le \tau \le \frac{T_r}{2} \tag{1}$$

The time pulse width, represented by $T_r$, and fast time $\tau$ represent the time variables in this radar system. The rate of chirp, denoted by $k_r = \pm\frac{B}{T_r}$, reflects the positive or negative rate based on the system requirements, where $B$ represents the bandwidth. The range resolution $\rho_r$ can generally be approximately expressed as

$$\rho_r \approx \frac{c}{2} \cdot \frac{1}{B} \tag{2}$$

where $c$ represents the velocity of light. To acquire a higher-resolution image, a wider bandwidth is required, which imposes substantial requirements on the DAC. Therefore, to effectively achieve a wideband LFM signal, the signal can be initially generated in the baseband and then modulated to the carrier frequency $f_0$ using a quadrature unit, resulting in Equation (1). The baseband signals $I(\tau)$ and $Q(\tau)$ can be expressed as follows:

$$\begin{cases} I(\tau) = \cos(\pi \cdot kr \cdot \tau^2) \\ Q(\tau) = \sin(\pi \cdot kr \cdot \tau^2) \end{cases} \tag{3}$$

The signal circuit generator typically consists of an FPGA and a DAC. Due to the nature of the FPGA, the LUT method can effectively implement the signal generator. As shown in Figure 1a, pre-calculated waveform data can be stored in FLASH in advance. Depending on various operational instructions, waveform data from different FLASH addresses can be retrieved and cached in RAM. This data are then transmitted to the DAC, which is controlled by the PRF.

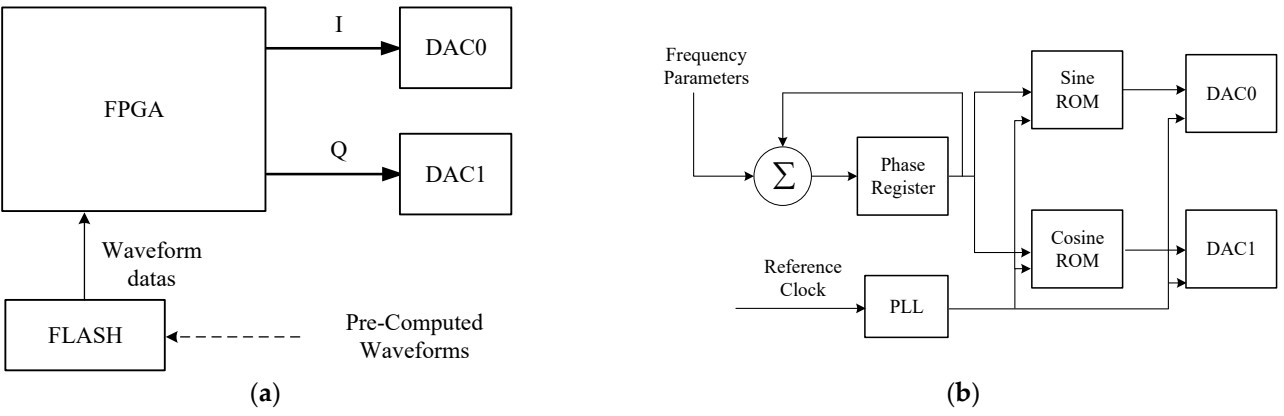

**Figure 1.** Common LFM signal generators: (**a**) the stored method [15]; (**b**) the DDS method [16].

The DDS method requires fewer storage resources compared to the storage method, as demonstrated in Figure 1b. This method accumulates phase information using frequency parameters, such as the reference clock and the length of the phase accumulator. It then uses this information as an address to read waveforms from ROM.

To generate various modes of the large TBP LFM for a high-resolution SAR system, the proposed approaches require waveform storage or relevant information, leading to significant storage resource consumption and inflexible implementation. In order to address the issue of extensive storage resource usage, a real-time waveform calculation method is suggested. This method, depicted in Figure 2, processes the signal using HLS IP and computes the signal's current value based on parameters such as sampling frequency and time pulse width, thereby expressing the instantaneous $\tau$ as follows:

$$\begin{cases} S = T_r \cdot F_s \\ \tau = (\tau_{in} - S/2)/F_s \end{cases} \tag{4}$$

where the value of $\tau_{in}$ is the digital counter's result with a step of 1, which ranges from 0 to $S - 1$, where $S$ represents the sample points. The following steps are caching and

parallelizing these serialized results, and upon completion, reading these parallel data from RAM.

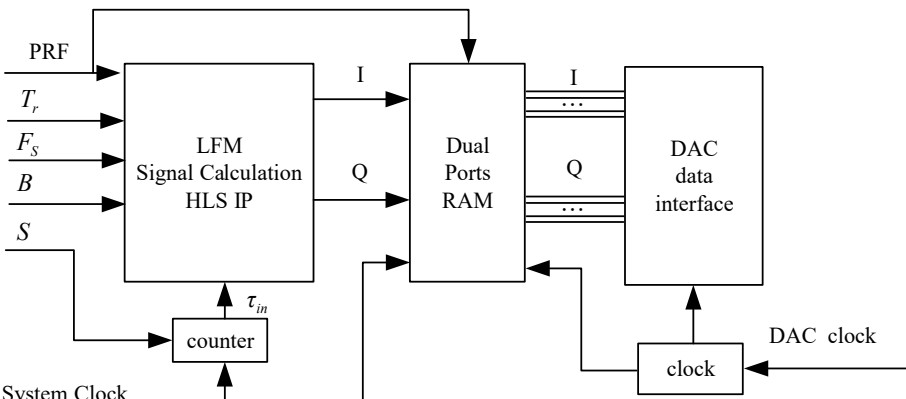

**Figure 2.** Block diagram of the real-time process. Compute serially and cache in parallel [18].

The flexible signal generation method described in this research is efficient in terms of storage resources, but its serial computation of the signal does not meet the requirements of the DAC's data path rate. Furthermore, the parallel implementation, especially when combined with predistortion, consumes excessive logic and DSP resources, potentially exceeding the limitations of the FPGA. Consequently, multiplexing this serial module becomes impractical. To address this issue, a high-speed dual-port RAM is necessary to cache the serially calculated results and parallelize the data. Subsequently, the parallelized waveforms can be read using the DAC clock, triggered by the PRF. The temporal correlation between a PRF pulse and the resultant signals is depicted in Figure 3. However, in some sophisticated SAR systems, there is a demand to generate different parameters, such as chirp rate, bandwidth, and time width, for the LFM signal in every PRF pulse. Given the time-consuming process of caching and parallelizing the results, the serial method cannot fulfill the timing requirements of these systems.

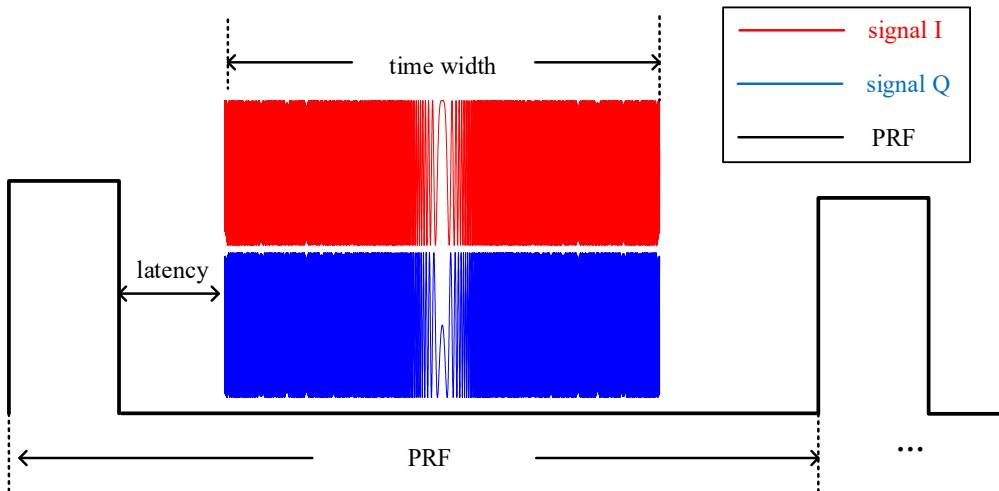

**Figure 3.** A graphical representation depicting the timing of the signals produced within a PRF pulse.

Therefore, in order to achieve the real-time capability of large TBP signals, a new and improved parallel calculation approach is necessary. Due to the complexity and cost of phase error compensation, this paper introduces a novel piecewise approximation algorithm. When integrated with the parallel signal calculation module, this proposed method can enable real-time processing of multipath LFM signal generation. More detailed explanations of the proposed algorithms will be provided in the following sections.

*2.2. Improved Parallel Real-Time LFM Generation*

To calculate the phase of LFM in parallel, when compared with Equation (4), the generation of instantaneous time $\tau$ needs to be modified as follows:

$$\tau(i) = (\tau'_{in} - \frac{S}{2})/F_s + i * \Delta\tau, \ i \in (0, N-1) \tag{5}$$

where $N$ represents the number of parallel data paths, simultaneously, and also the step of the digital counter; and $\tau'_{in}$ is the result of the counter from 1 to $S$. Meanwhile, after acquiring $\tau(0)$, the result of other paths $\tau(i)$ can be calculated via addition, where the difference in the adjacent value $\Delta\tau$ is equal to $1/F_s$. Compared with multiplexing serial calculation modules, the instantaneous time can be computed in parallel. This method can efficiently reduce the consumption of resources and optimize through the pipeline.

To determine the phase errors of the system in a closed-loop test mode, sample the demodulated I/Q signals from the RF circuit through ADC, process these data in the FPGA, and transmit these results to a digital recorder. The phase of echo waves can be obtained using the unwrap function. Then, calculate the mean value of these phase data and subtract the ideal reference phase. The phase error can be expressed as

$$\phi_{err}(\tau) = \frac{1}{M}\sum_{i=1}^{M}(\phi_i(\tau) - \phi_{ref}(\tau)) \tag{6}$$

where $\phi(\tau)$ represents the phase of the echo wave, $\phi_{ref}(\tau)$ represents the phase of the ideal signal with identical parameters, and $M$ represents the number of PRF pulses. The polynomial curve is generally utilized to illustrate the phase errors. The formula for a $k - order$ polynomial can be expressed as:

$$f(\tau) = \sum_{i=0}^{k} h_i \cdot (\tau)^i + o(\tau^k) \tag{7}$$

where the parameters $h_0, h_1, h_2, h_3 \cdots h_k$ represent the coefficients of a $k - order$ polynomial curve, and $o(\tau^k)$ represents approximation error. Once these parameters have been obtained, the phase errors can be expressed as:

$$\phi_{err}(\tau) = h_0 + h_1 \cdot \tau + h_2 \cdot \tau^2 + \cdots h_k \cdot \tau^k \approx f(\tau) \tag{8}$$

Due to the complexity of the phase error, real-time calculation of high-order polynomials is necessary to approximate the curve, which requires significant logic and DSP resources. Consequently, this function cannot be reused in parallel across multiple paths. When combined with phase calculation, the phase expression is better suited for serial computation. To achieve parallel calculation within limited resources, the calculation algorithm needs to be enhanced. The specifics of the proposed algorithm will be presented in the next chapter.

*2.3. Improved Parallel Polynomial Calculation Algorithm*

In order to perform parallel calculations of polynomial expressions with limited resources, this paper proposes a piecewise linear approximation of the polynomial curve The difference between consecutive results is determined by the step size, which is based on the number of parallel paths $N$. For instance, in Figure 4a, a $3 - order$ polynomial curve is divided into $S/N$ portions, while in Figure 4b, the division unit can be replaced by a linear

segment with a slope defined by adjacent values and parallel paths $N$. The expression of the piecewise phase error curve can be presented as follows:

$$
\begin{cases}
\phi_{err}(0) + \dfrac{\phi_{err}(N) - \phi_{err}(0)}{N} \cdot i, \ 0 \le i \le N-1 \\[2mm]
\phi_{err}(N) + \dfrac{\phi_{err}(2*N) - \phi_{err}(N)}{N} \cdot i, \ 0 \le i \le N-1 \\[1mm]
\cdots\cdots \\[1mm]
\phi_{err}(S-N) + \dfrac{\phi_{err}(S) - \phi_{err}(S-N)}{N} \cdot i, \ 0 \le i \le N-1
\end{cases}
\tag{9}
$$

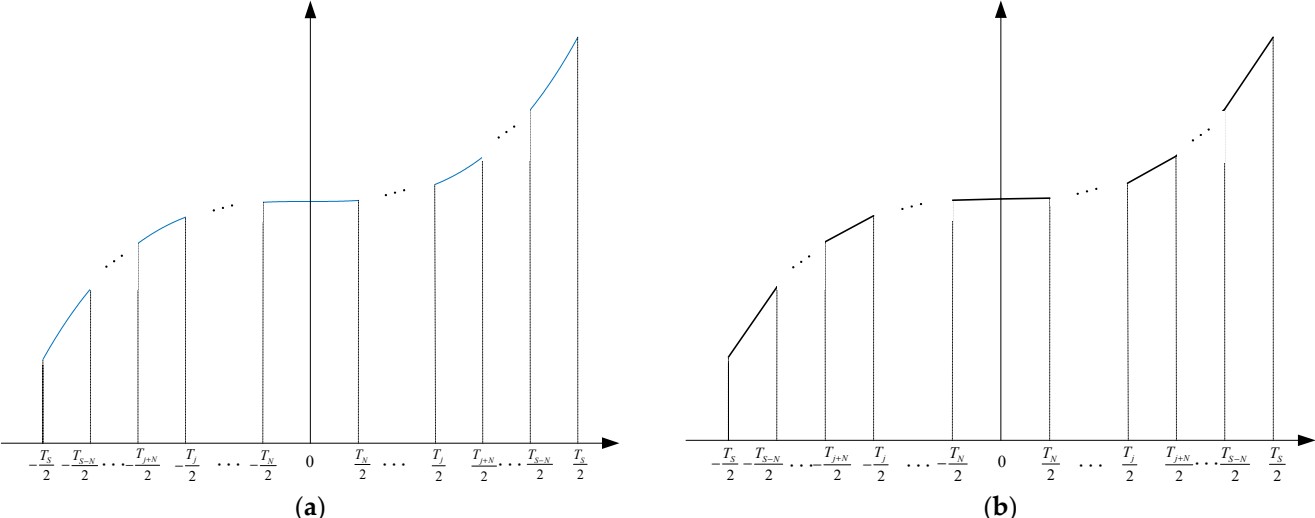

**Figure 4.** An example of a $3-order$ polynomial curve and piecewise linear, where $j = 0, N, 2N, \cdots,$ $S-N, S$: (**a**) a $3-order$ polynomial curve divided by $N$; (**b**) the division curve's piecewise linear approximation.

The slope of a piecewise line can be determined by using the neighboring values based on the given formulas. In order to maintain continuity at the boundaries of piecewise linear approximations, $\phi_{err}(j), (j = 0, N, 2N, \cdots, S-N, S)$ should be computed. Calculating the difference in $\phi_{err}(j)$ requires the calculation of adjacent values, which consumes double logic and DSP resources. In this regard, taking into account the characteristics of FPGA, the improved method can compute the polynomial and cache the result in the register. Subsequently, the current value is subtracted from the registered value to obtain the difference, which can be optimized through pipelining during implementation. The expressions can be simplified as follows:

$$
\begin{cases}
\phi_{err}(N)\_reg + \dfrac{\phi_{err}(N) - \phi_{err}(N)\_reg}{N} \cdot i, \ 0 \le i \le N-1 \\[2mm]
\phi_{err}(2N)\_reg + \dfrac{\phi_{err}(2N) - \phi_{err}(2N)\_reg}{N} \cdot i, \ 0 \le i \le N-1 \\[1mm]
\cdots \\[1mm]
\phi_{err}(S)\_reg + \dfrac{\phi_{err}(S) - \phi_{err}(S)\_reg}{N} \cdot i, \ 0 \le i \le N-1
\end{cases}
\tag{10}
$$

### 2.4. Improved Parallel Real-Time LFM Signal Generation Method

Based on the aforementioned considerations, this paper introduces a novel real-time parallel LFM generator that is capable of calculating signals in parallel while conserving resources. To concurrently calculate the predistortion phase of the signal using Formula (10), it is necessary to compute the instantaneous time $\tau$ based on Equation (4). Moreover, to guarantee continuity at the edge of the piecewise linear phase, the $S/N$ numbers of

$\tau(j), (j = 0, N, 2N, \cdots, S - N, S)$ should be calculated. Then, $\phi_{err}(j), (j = 0, N, 2N, \cdots, S - N, S)$ can be determined according to Equation (8), where $h_i$ can be acquired from the instructions or stored in the program. The acquired $\phi_{err}(j)$ and $\tau(j)$ can be used to cache these values in registers. Afterwards, the phase $\theta(\tau)$ of the LFM signal can be calculated in parallel, and $\theta(\tau)$ can be expressed as follows:

$$
\begin{cases}
\tau(j) = (\tau\_in(j) - \dfrac{S}{2})/F_S, \ j = 0, N, 2N, \cdots, S - N, S \\[2mm]
\Delta\phi_{err}(j) = \dfrac{\phi_{err}(j) - \phi_{err}(j)\_reg}{N}, \ j = 0, N, 2N, \cdots, S - N, S \\[2mm]
\Delta\tau = 1/F_S \\[2mm]
\theta(\tau) = \pi \cdot kr \cdot (\tau\_{reg}(j) + \Delta\tau \cdot i)^2 - (\phi_{err}(j)\_{reg} + \Delta\phi_{err}(j) \cdot i), \ 0 \le i \le N-1
\end{cases}
\tag{11}
$$

where $\tau\_in$ represents the digital counter with an incrementation step of $N$.

Therefore, the $N$ paths of the LFM signal phase can be concurrently computed in real-time without requiring additional logic and DSP resources. Following the phase calculation, the waveform can be obtained using cosine and sine functions. These complex calculations can be realized in HLS IP to simplify the complexity of implementation. In Figure 5, the block diagram demonstrates the implementation of complex computations in HLS IP. Simultaneously, when combined with the characteristics of FPGA, utilizing registers and pipelines can achieve parallel computation through piecewise linear approximation. Consequently, this method of waveform generation can execute advanced signal processing in real-time.

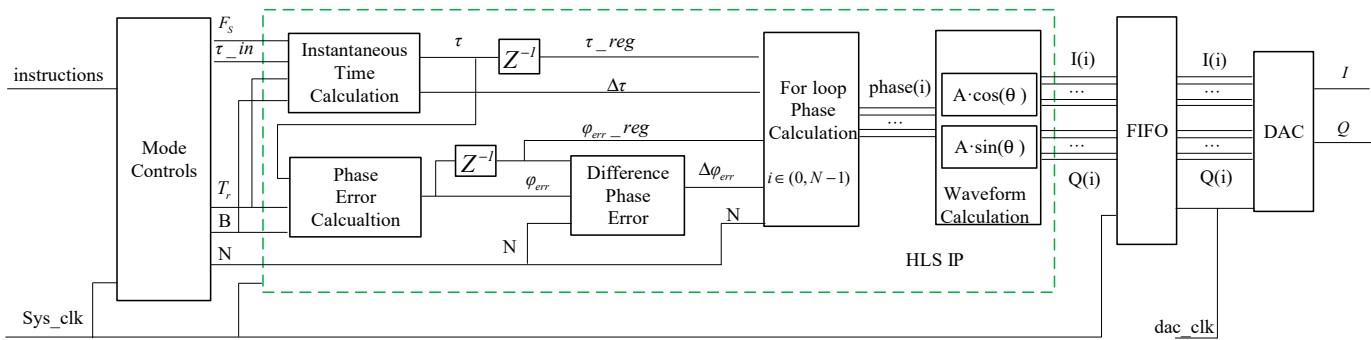

**Figure 5.** Block diagram of parallel signal generation in real-time. The circuits within the green dotted box represent the HLS IP.

To validate the performance of the proposed algorithm, experiments were conducted using an X-band SAR system. The parameters of the experimental signal are detailed in Table 1. The proposed algorithm was processed in a digital synthesis circuit, mainly consisting of FPGAs, DACs, SoC, a clock chip, and DACs. As depicted in Figure 6a, the LFM signal generator was verified using FPGA1 and dual-channel DAC1, while echo wave sampling was based on FPGA2 and dual-channel ADC1. The SoC controls and monitors the SAR system. In the closed-loop test mode, as depicted in Figure 6b, the generated I/Q signals are modulated to the X-band in the RF circuit, and then filtered by a band-pass filter (BPF) and amplified by a power amplifier (PA). This full-power signal is radiated into free space by a waveguide slot antenna. The transmitted signal is received by the horn antenna and then passed from circulator port 1 to circulator port 2. It is then transmitted to an optical delayer, which controls the latency time to simulate a real working environment. The delayed signal is conveyed to the horn antenna via circulator port 3 to circulator port 1, and the SAR antenna receives the transmitted signal from the horn antenna. The received signal is filtered by a BPF and amplified by a linear amplifier (LNA). Subsequently, this signal is demodulated into I/Q and sampled by the ADC. After processing the sampled data, it is transmitted to the digital recorder via optical fiber transmission.

**Table 1.** Parameters of experimental signal.

| Parameter | Value |
| --- | --- |
| DAC sampling frequency | 2 GHz |
| Bandwidth | 440 MHz |
| Time width | 120 μs |
| Carrier frequency | 9.6 GHz |
| Pulse repetition frequency (PRF) | 600 Hz |
| ADC sampling frequency | 2000 MHz |
| Sampling points | 81,920 |

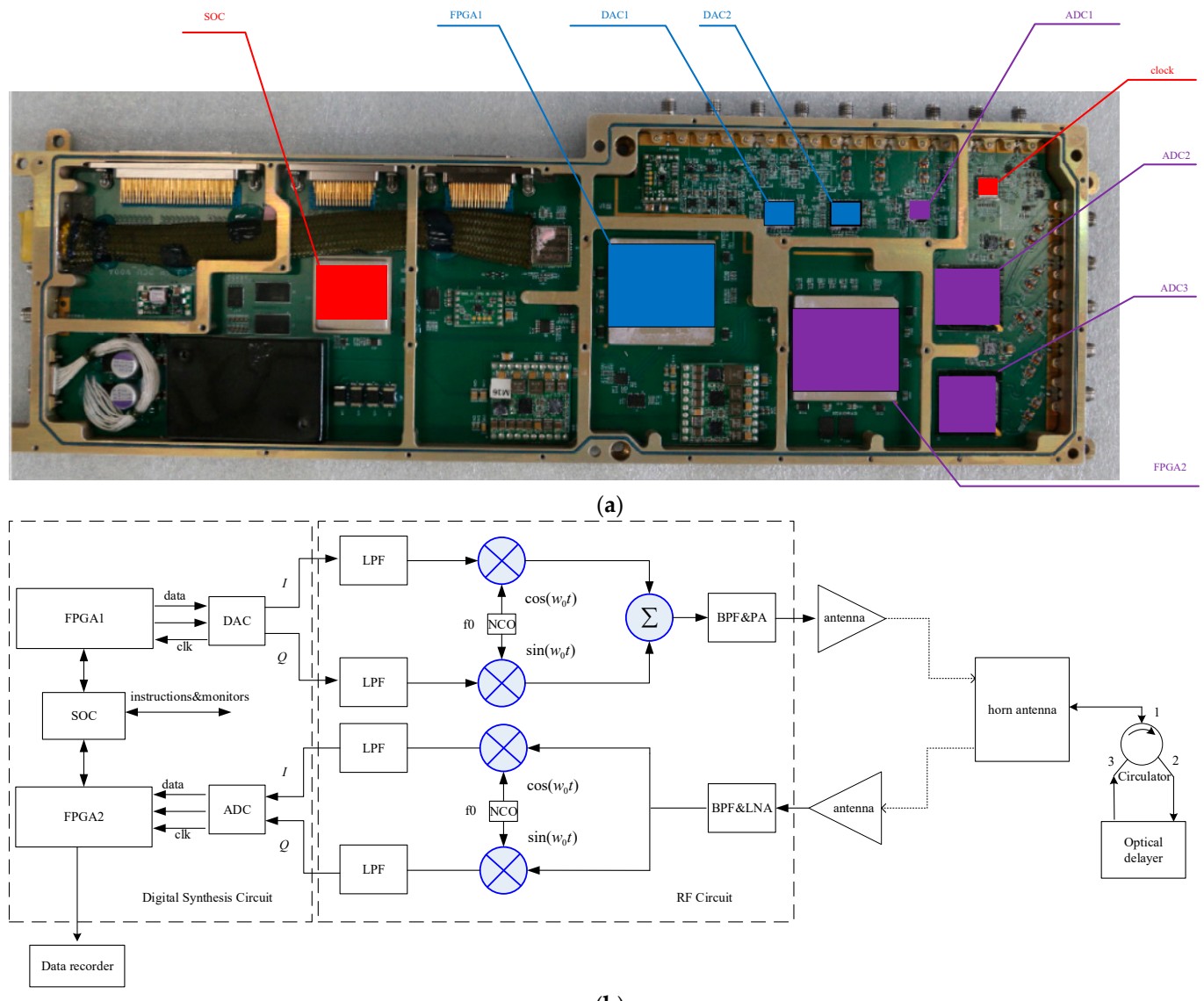

(**a**)

(**b**)

**Figure 6.** Experiment platform: (**a**) digital synthesis circuit; (**b**) block diagram of closed-loop test mode.

## 3. Results

### 3.1. Results of the LFM Signal Generation

At first, the I and Q signals are acquired by a high-speed oscilloscope, with waveforms displayed in Figure 7a. The generated I and Q signals exhibit smoothness, consistent amplitude, and orthogonal phase, with a pulse duration of 120 μs (each unit represents 20 μs on the X-axis).

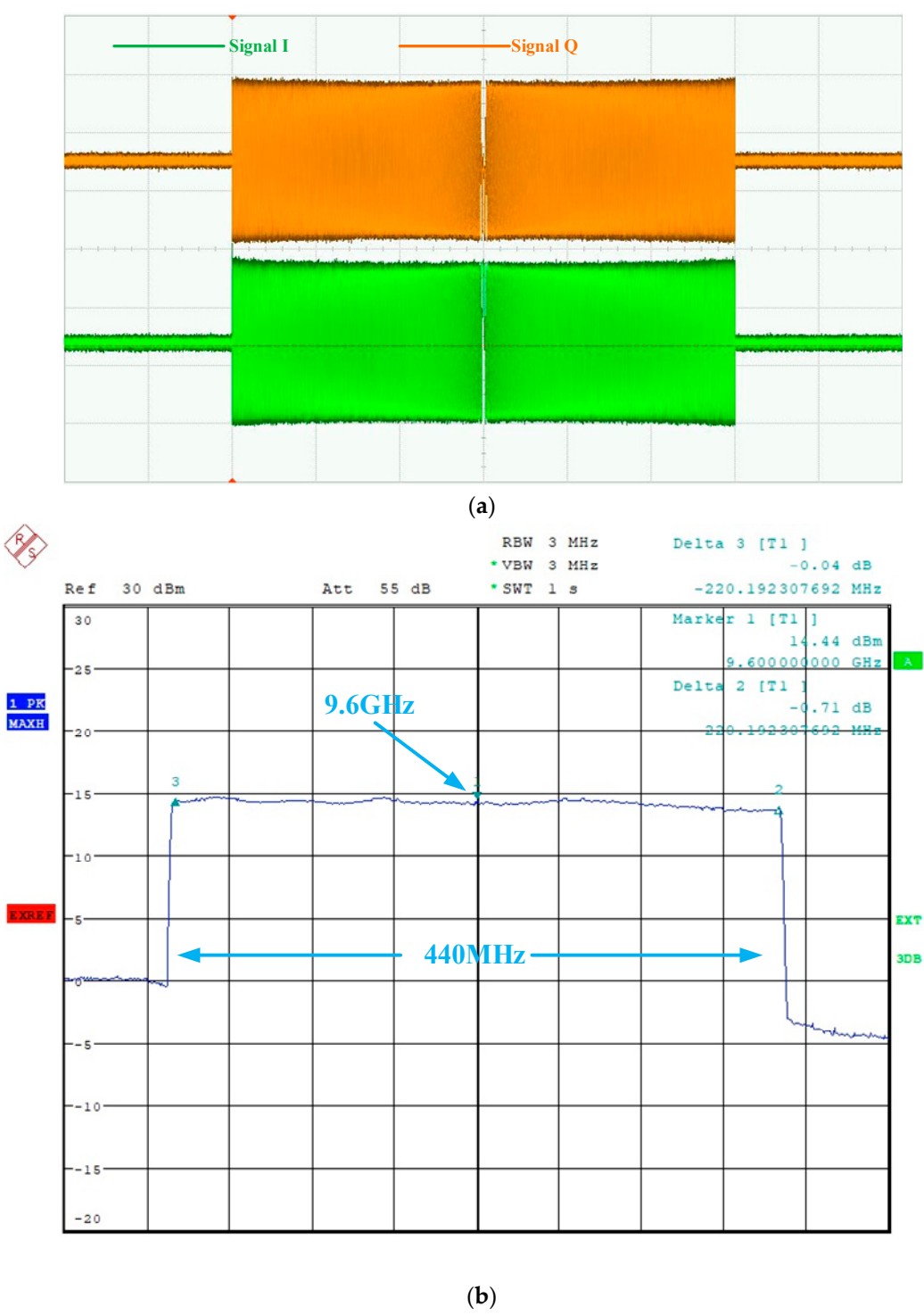

**Figure 7.** The LFM signal has a bandwidth of 440 MHz and a time width of 120 μs: (**a**) the I and Q signal waveforms, captured by the high-speed oscilloscope, have a time width of 120 μs; (**b**) the spectrum of the modulated signal shows a center frequency of 9.6 GHz and a bandwidth of 440 MHz.

Subsequently, the I and Q signals are modulated to a carrier frequency of 9.6 GHz in the RF circuit. As depicted in Figure 7b, the frequency spectrum of this signal ranges from 9.38 GHz to 9.82 GHz, yielding a bandwidth of 440 MHz. Given that the DAC's sampling rate is 2 Gsps and the data path count is 16, with I and Q occupying 8 lanes each at a rate of 250 Msps per lane, a calculation clock of 125 MHz and a parallel number *N* of 16 are required to meet this rate demand. Then, the generator can realize real-time

processing. Here, the waveforms and the frequency spectrum of this signal serve to validate the performance of the proposed parallel computation algorithm.

### 3.2. Results of Phase Compensation

In the closed-loop test mode of the SAR system, the echo data are recorded by the dedicated test apparatus. The recorded waveform $I(\tau)$ and $Q(\tau)$ for one PRF pulse are shown in Figure 8a. To acquire the pulse compression result $Comp(\tau)$ of this signal, matched filtering should be implemented. The ideal reference signal $chirp\_ref$ can be expressed as

$$\begin{cases} f_r = ((0:n_r-1) - \frac{n_r}{2}) \cdot F_s/n_r \\ chirp\_ref(f_r) = \exp(-i \cdot \pi \cdot \frac{f_r^2}{kr}) \end{cases} \tag{12}$$

where $n_r$ represents the sampling points. Pulse compression can be achieved by using the $fft$ and $ifft$ functions, and $Comp(\tau)$ can be expressed as

$$\begin{cases} chirp\_echo(\tau) = I(\tau) + i \cdot Q(\tau) \\ Comp(\tau) = ifft(fft(chirp\_echo(\tau)) \cdot chirp\_ref(f_r)) \end{cases} \tag{13}$$

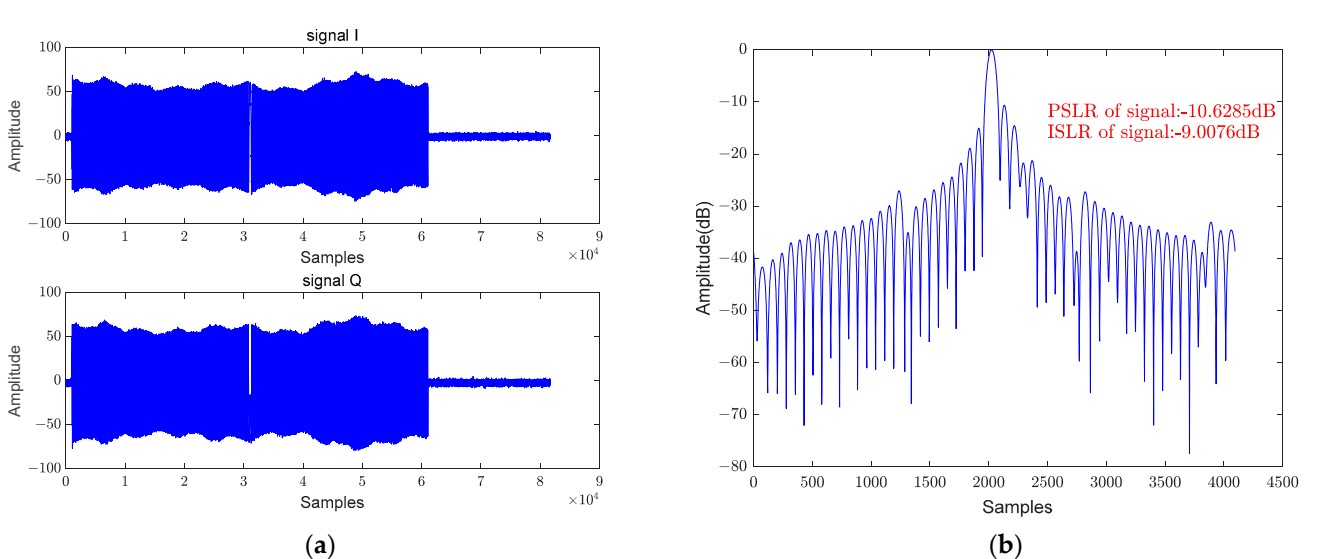

**Figure 8.** Processed signal from closed-loop test mode: (**a**) the closed-loop recorded waveforms for signal I and signal Q; (**b**) the pulse compression result with a bandwidth of 440 MHz and a time width of 120 μs.

To visually analyze the test results, interpolation was applied to the pulse compression results of all test signals. Specifically, 64 points were selected at the peak absolute value of the pulse compression results and then interpolated 64 times. Subsequently, PSLR and ISLR could be calculated from the interpolated results. As shown in Figure 8b, the PSLR of the recorded waveform was −10.6285 dB, and the ISLR of the recorded waveform was −9.0076 dB.

As illustrated in Figure 9a, the system's phase errors were captured by calculating the phase errors of the recorded signals in comparison with the ideal reference signal, averaging the results over 50 PRF pulses. Furthermore, a $14-order$ polynomial curve was employed to approximate this phase error. To calculate this phase error in parallel, the polynomial curve could be segmented into multiple parts, determined by the number of data paths $N$. In this research, in order to demonstrate the proposed parallel algorithm, different parameters⁻$N = 4$, $N = 8$, and $N = 16$—were validated. The piecewise linear divisions corresponding to these parameters are depicted in Figure 9b. In contrast with the polynomial curve, a difference of 0.1 was added to each piecewise linear segment. As

depicted in Figure 9b, the continuity at the edge of each division is ensured by the proposed piecewise linear curve.

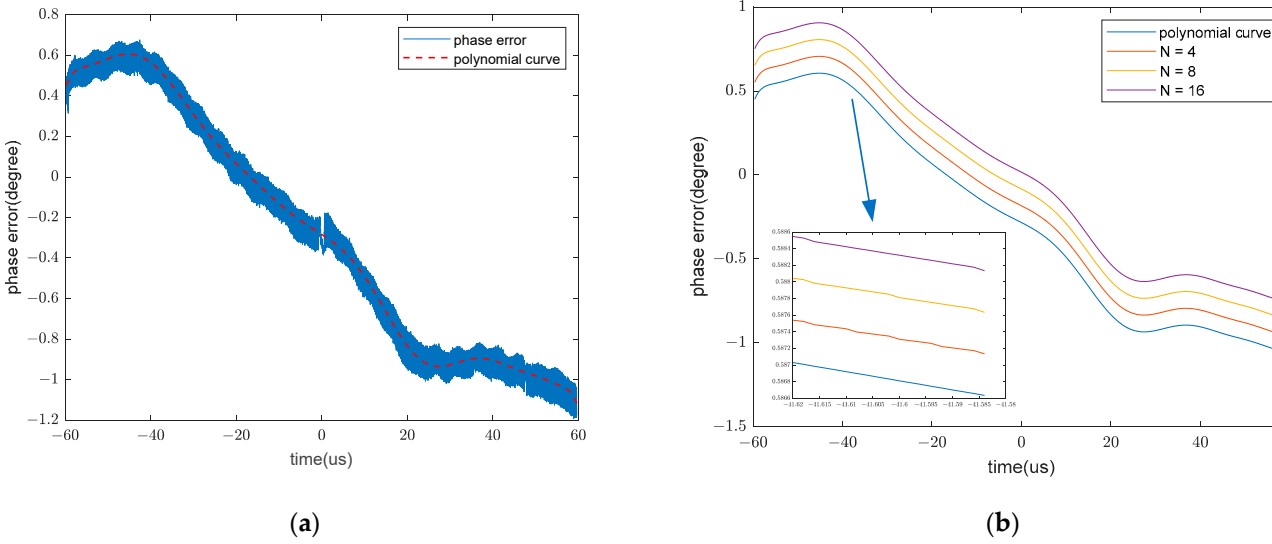

**Figure 9.** The phase error of the system and approximate curves: (**a**) the mean phase error derived from 50 pulses and a 14-order polynomial curve; (**b**) piecewise linear divisions for different parameters of $N$.

After acquiring various approximation curves and offsetting the phase errors in the signal, the compensation effects of different curves could be determined. The results of pulse compensation are depicted in Figure 10. As shown in Figure 10a, when compensated with the polynomial curve, the PSLR was $-14.7659$ dB and the ISLR was $-11.1263$ dB. As depicted in Figure 10b, when compensation was applied using an $N = 4$ piecewise linear curve, the PSLR was $-14.7661$ dB and the ISLR remained at $-11.1263$ dB. Similarly, in Figure 10c,d, it can be seen that compensating with $N = 8$ and $N = 16$ piecewise linear curves both yielded a PSLR of $-14.7662$ dB and an ISLR of $-11.1263$ dB. Given the nearly identical results of these compensated signals, it can be concluded that the application of piecewise linear curves with different $N$ values achieves the same effect as the polynomial curve.

### 3.3. Results of Signal with Compensation

After acquiring the coefficients of the $14 - order$ polynomial formula, the parallel calculation expressions could be acquired, and the proposed compensation algorithms could be implemented in the digital synthesis hardware using the HLS method. To validate the effectiveness of the compensation, we tested it in the closed-loop mode. As depicted in Figure 11a, the recorded waveforms of I and Q were accurate, with the PSLR and ISLR of the recorded data at $-13.0113$ dB and $-10.746$ dB, respectively, as depicted in Figure 11b, which were nearly identical to the ideal signal.

The results from the simulation and implementation of the proposed generator validate the possibility of parallel real-time computation when faced with constrained logic and DSP resources. This approach allows for the real-time computation of complex signals, which is particularly advantageous for specific applications that demand the generation of signals with varying parameters in adjacent PRFs. In contrast, the storage method and serial calculation method do not meet the time-based requirements. By overcoming their limitations, this parallel processing implementation can effectively generate an LFM signal with varying parameters, including bandwidth, time width, and chirp rate, in every PRF pulse.

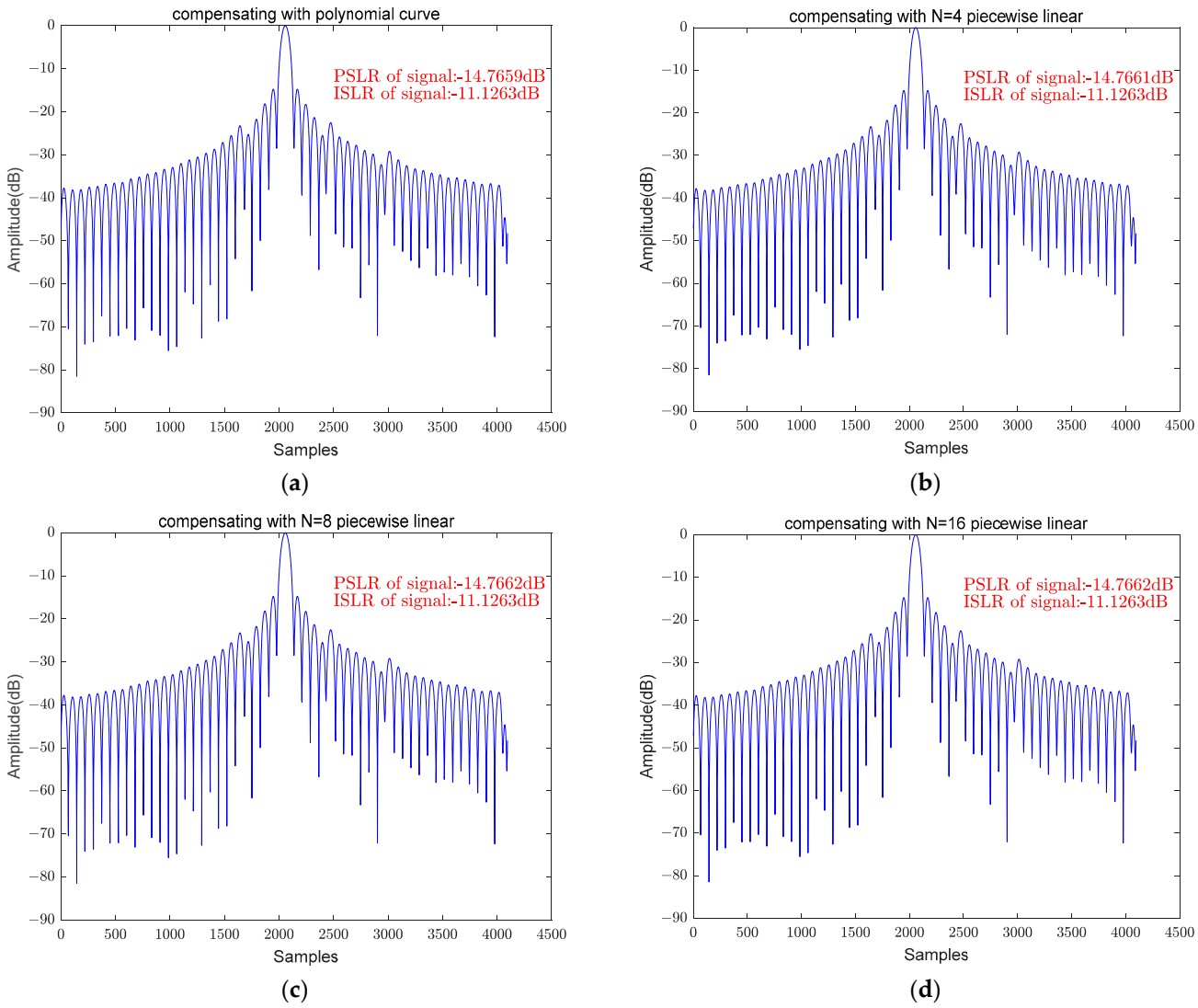

**Figure 10.** The results of pulse compression with compensation: (**a**) The result of pulse compression when compensated with the polynomial curve. The result of pulse compression when compensated with an (**b**) *N* = 4, (**c**) *N* = 8, or (**d**) *N* = 16 piecewise linear curve.

### 3.4. Comparison of Different Methods

3.4.1. Comparison of Indicators

In order to visually observe the test results more effectively, interpolation was applied to the pulse pressure results of the three modes. We selected 32 points at the maximum absolute value of the pulse pressure results, and 128-fold interpolation was used. The PSLR and ISLR were calculated based on these interpolated results. Simultaneously, based on the same parameters, a 16-bit signed ideal reference signal was generated. The performance indicators of the three waveform generation methods, namely the storage method, serial real-time computing method, and parallel real-time computing method, were compared with the ideal reference signal on the same hardware platform, and the results are presented in Table 2.

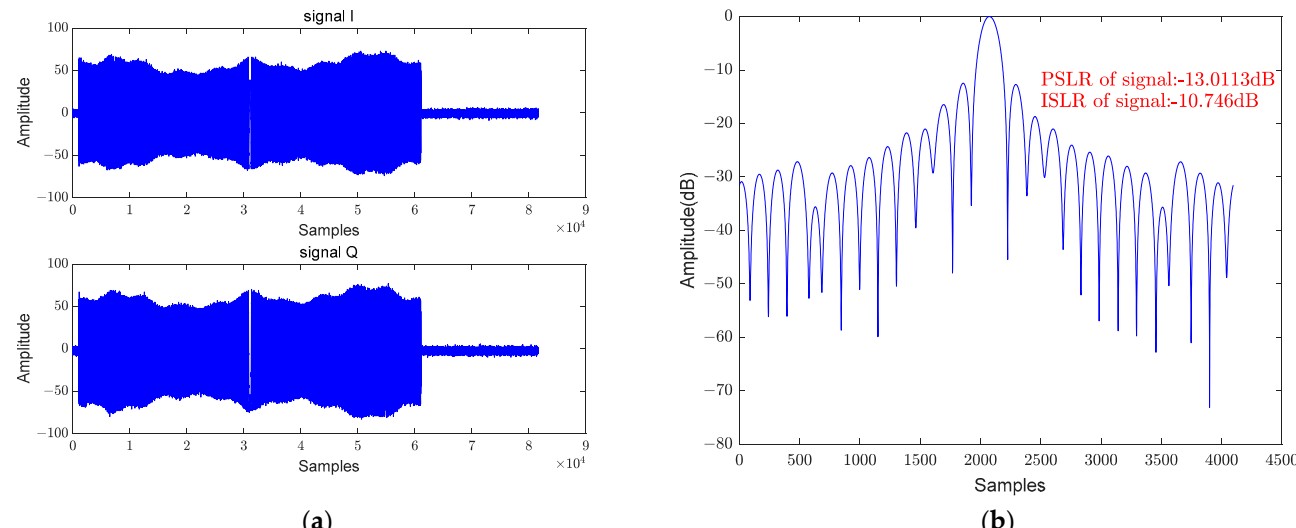

**Figure 11.** The signals with predistortion: (**a**) recorded waveforms of signal I and signal Q; (**b**) the result of pulse compression with predistortion.

**Table 2.** The comparison of PSLR and ISLR in different methods.

| Method | Storage Method [15] | Serial Calculation Method [18] | Parallel Calculation Method |
|---|---|---|---|
| PSLR of ideal signal (dB) | −13.2592 | −13.2592 | −13.2592 |
| ISLR of ideal signal (dB) | −10.1038 | −10.1038 | −10.1038 |
| PSLR of signal without compensation (dB) | −10.5589 | −10.5613 | −10.6285 |
| ISLR of signal without compensation (dB) | −9.0688 | −9.0698 | −9.0096 |
| PSLR of signal with compensation (dB) | −13.0051 | −13.0138 | −13.0113 |
| ISLR of signal with compensation (dB) | −10.747 | −10.748 | −10.746 |

### 3.4.2. Comparison of Resource Utilization

Based on the comparison of the test results of the three methods, we can surmise that the qualities of the signals generated by the three methods were very close when tested on the same hardware platform. However, the three methods required different hardware resources and processing times. The storage method relied on a substantial amount of dedicated hardware storage resources, whereas serial real-time computation and parallel real-time computation did not. The required storage resources for a 16-bit dual-channel DAC can be computed using the following equation:

$$F = 2 \times 2 \times F_S \times \sum_{1}^{n} T_r(i) \tag{14}$$

where $T_r(i)$ represents the time width of the $n$ various parameter signals. For example, in the case of the 120 μs signal from the experiment, it would require 937.5 KB of storage resources to store the signal. Therefore, a system consisting of multiple modes with large time widths would require a substantial quantity of storage resources for waveform storage. Furthermore, when replacing waveform data, it is necessary to erase and reprogram the FLASH chip, which is very inflexible for system implementation. In contrast, neither the serial nor the parallel real-time processing method requires dedicated hardware storage resources.

As a point of comparison, the time required for these methods is another crucial metric that needs to be measured and evaluated. The storage method involves a significant

amount of time to read the saved waveforms from the FLASH chip initially. Subsequently, the data are cached and parallelized through a dual-port RAM inside the FPGA. Finally, the parallelized waveform can be acquired using the DAC clock. Meanwhile, due to the rate mismatch between the DAC's data path and waveform calculation, the serial real-time method requires additional time to cache and parallelize the signal in a dual-port RAM after completing the waveform processing. In contrast, the parallel real-time method only requires the time needed for signal processing delay, without the need for additional caching and parallelization of the signal. A comparison of time and resource consumption for the three methods is shown in Table 3.

**Table 3.** Comparison of hardware resources and implementation time.

| Method | Storage Method | Serial Calculation Method | Parallel Calculation Method |
|---|---|---|---|
| The dedicated hardware storage resources (KB) | 937.5 | 0 | 0 |
| The dedicated BRAM cache in FPGA | 14.4% | 14.4% | 0.4% |
| The time necessary to prepare the waveform for processing | 38.5 ms | 2.5 ms | 1.7 μs |
| The utilization of DSP resources in FPGA | 0 | 33% | 39% |
| The utilization of LUT resources in FPGA | 9% | 23% | 26% |

In addition, there is a significant discrepancy in the internal consumption of LUT and DSP resources within the FPGA for the three approaches. Due to the waveforms being stored in FLASH when using the storage method, this approach does not demand a substantial amount of logical resources. Conversely, the real-time methods consume a significant portion of these resources. A comparison of LUT and DSP resource consumption for the three methods is illustrated in Table 3. As shown in Table 3, the utilization of LUT and DSP resources in the serial real-time method is significant. Even though multiplexing this module $N$ times can be optimized, it would still surpass the limitations of the FPGA.

When comparing these methods, it becomes evident that each approach offers distinct advantages and comparable performance on the same hardware platform. The storage method can be relatively straightforward to implement, but it necessitates a dedicated FLASH for storing waveforms of multiple modes of large TBP signals. It also requires a significant amount of time for preparation before transmitting to DAC. On the other hand, the serial processing method enables real-time signal calculation and is more adaptable than the storage method, yet it requires substantial DSP and LUT resources. This limitation prevents the module from multiplexing and necessitates time for caching and parallelizing data. Additionally, this method fails to meet the time requirements for systems with different parameter signals in adjacent PRFs. Therefore, when searching for a multifunctional airborne SAR system that needs to be adaptable in various complicated scenarios, the parallel real-time calculation method emerges as the optimal approach for implementation.

## 4. Discussion

As a critical tool for remote sensing, the airborne SAR system operates in various conditions, incorporating multiple functions that require different signal modes. Additionally, the system places significant demands on signal generation in complex electromagnetic environments. In specific RF stealth scenarios, it may be necessary to generate signals with different parameters in adjacent PRFs. In HRWS SAR systems, multiple operating modes consist of a variety of bandwidth and time width parameters. There is a requirement for a more advanced generation approach to replace the stored method, which not only consumes substantial storage resources but also offers limited flexibility. Given these limitations, it is more appropriate to process LFM signals within an FPGA in real-time, relying

simply on crucial parameters such as bandwidth, time width, chirp rate, and sampling frequency. Simultaneously, it is essential to implement predistortion in the signal generator to compensate for the phase errors of SAR. These errors can be approximated using high-order polynomial curves. The HLS method can simplify this complex computation more effectively than traditional RTL design. Given the data rate requirement of high-speed DAC, which necessitates the parallel transmission of multipath data, it is impractical to multiplex the calculation module with predistortion in an FPGA with limited resources. Consequently, a serial calculation method and the parallelization of waveform data in dual-port RAM are proposed. While the serial computation method can process signals flexibly without storing any data, it does require some time to cache the serial calculations in RAM. Subsequently, it reads the parallelized data from the RAM to the high-speed DAC.

Due to the anti-jamming requirements in some complicated electromagnetic environments, these might demand that signals are generated with different parameters in contiguous PRF pulses. Therefore, to satisfy the time-sensitive requirements of a high-resolution SAR system, parallel computing is essential for implementing the large TBP signal generation. In this paper, we have proposed a novel parallel piecewise linear algorithm as an alternative to multiplexing serial calculation modules, which can efficiently conserve logic and DSP resources. This proposed method enables real-time signal generation, combined with the system's sequential and parametric controls, allowing for different waveforms between two adjacent PRFs.

This paper has compared the storage method, serial calculation, and the proposed parallel calculation on the same experimental platform. While all these methods generated high-quality wideband LFM signals, only the parallel calculation could produce LFM signals with different parameters in adjacent PRFs. With this finding, this paper paves the way for further research into anti-jamming applications.

## 5. Conclusions

In this paper, we have introduced a novel parallel real-time LFM signal generator. Unlike conventional methods that rely on pre-stored waveform or phase information, the real-time method processes signals based solely on crucial parameters. The challenge with large TBP signals lies in the impracticality of multiplexing the serial calculation module within the limited resources of an FPGA. Consequently, this necessitates high-speed dual-port RAM to cache the waveforms, which impacts the timeliness of the generated signals. In contrast, the proposed algorithm utilizes piecewise linear computations in parallel to replace multiplexing polynomial calculations. This proposed method can calculate predistorted signals with varying parameters in real-time, even between adjacent PRFs, making it particularly suitable for complex electromagnetic environments in future applications. The effectiveness of the proposed algorithms has been verified with the results of simulations and experiments.

**Author Contributions:** Conceptualization, D.C. and T.W.; methodology, D.C., T.W., J.F. and X.Y.; software, D.C. and X.Y.; validation, D.C., X.Y., G.L. and J.Z.; formal analysis, D.C., J.F. and Z.Y.; investigation, D.C., T.W., J.F. and Z.Y.; writing original draft preparation, D.C. All authors have read and agreed to the published version of the manuscript.

**Funding:** This research received no external funding.

**Data Availability Statement:** Dataset available on request from the authors.

**Acknowledgments:** We would like to thank the School of Microelectronics, Northwestern Polytechnical University, China; the Aerospace Information Research Institute, Chinese Academy of Sciences, China; and the School of Electronic, Electrical and Communication Engineering, University of Chinese Academy of Sciences.

**Conflicts of Interest:** The authors declare no conflict of interest.

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
