# Peer review of "A Novel Real-Time Processing Wideband Waveform Generator of Airborne Synthetic Aperture Radar"

_remotesensing, doi:10.3390/rs16030496_

Round 1
Reviewer 1 Report
Comments and Suggestions for Authors
The paper is of certain interest, , it will be appreciated by readers positively. What is important, the authors presented the hardware implementation of their approach, the results of LFM signal generation, and gave an assessment of the signal quality in the form of the PSLR and ISLR level. But the article is devoted to the instrumental issues, and it seems to me that it may be somewhat off the Remote Sensing topic. Anyways, the paper may be accepted for publication provided some remarks below are discussed.
1. What is the role of PRF mentioned in the line 161? May be the successive pulses have different phase functions?
2. I think that for the better understanding of the phase function calculation from N pieces generated in parallel it would be desirable to write general expression for the target phase function instead of, say, the set of equations in (9). Otherwise it is really difficult to understand the way authors combine the pieces of the phase function.
3. How the problem of the phase discontinuity at the boundaries of piecewise linear approximations is solved if the pieces of the phase function are calculated separately and in parallel?
4. Symbol S in eq. 4 is in fact a number of the LFM pulse samples. Is it correct that in the number of equations in (9) is S/N? If true, that should be mentioned in the text clearly.
5. In the numerator within the brackets of the last equation of (9) there should be fi_err(S), not fi_err(N).
6. How authors got PSLR of ideal signal to be -13.2592 and ISLR -10.1038 in the Table 2? In the case of sinc squared function the PSLR is -13.465 and ISLR = -10.26.
Comments on the Quality of English LanguageQuality of English language is quite good, but some minor corrections are required.
Reviewer 2 Report
Comments and Suggestions for Authors
The paper introduces a method which generates parallel real-time LFM signal generator in a FPGA using HLS. Parallelization of the process in FPGA is described to achieve the results from the proposed method. Validation of the proposed method is carried out through simulations and experiments. The paper also talks about the predistortion for the signal using phase errors which is obtained using approximation of polynomial curve.
The paper fits the scope of the journal. I would recommend accepting the paper after the authors address the minor comments below:
Please go through the paper in detail for grammatical errors and sentence formations. Examples: Line 164- rewrite the sentence for clarity. Line 328 – it’’s to it’s.
The numbers in table 2 for LUT method, serial calculation method and parallel computation method are very close, comments on why this is observed can be added in the description.
Comments on the Quality of English LanguagePlease go through the paper in detail and look for grammatical errors and sentence formations.
Reviewer 3 Report
Comments and Suggestions for Authors
Dear Authors,
It was a pleasure to read such a good article. The proposed method is very interesting and possibly can be used very widely because resource availability is always an issue in all signal processing applications.
The article is written on a very high level, scientifically sound, and interesting for the reader.
I would especially underline the high quality of the abstract which provides necessary information in a compressed and digestible manner.
Specific comments.
1. Line 130. Probably "." is needed instead of "," before "to acquire". The whole sentence is confusing in its present form.
2. Currently, Figure 3 captions are on a separate page from Figure 3 itself. Would be nice if Authors or Editors take care of that.
3. Please make all the text and numbers in Figures 6a and 6b much larger.
4. The performance advantage of the Newly Proposed Parallel Calculation method over previous methods should be presented in anumerical forms. The authors claimed that the method decreases the resources (FPGA) needed for signal processing but that should be supported by some kind of simulation, which would show time reduction, or resource reduction (processing power or power consumption) if the new method is used. Please find the most appropriate metrics. Table 2 would be a good place to show that difference. Currently, all the numbers there are similar. The new result should be added to Results, Conclusions, and Abstract. If it has real-life importance, it would be a good point to add to the Discussion. Methods Section should also describe how that measurement was achieved.
Despite what was said in Comment #4, the Results of the Article are trustworthy and the quality of signal processing achieved by the method is presented very well.
5. This is the comment for your consideration. Currently article has a low number of references and they are mostly related to Signal processing. The Introduction briefly mentions the SAR usage for Earth observation. I believe that the article considers how the method might be used in practical realization for example on SAR earth observation satellites, it would have some benefits such as more images acquired per orbit, for example. What I'm trying to say is that if the article provided a real-life value of the method, it would improve the usefulness of the article even further.
I'm going to ask the article for a Major review but this is not because of the flaws of the article (which I don't see any), but because I believe that the article has a lot of additional potential, which can be used.
Thank you and good luck with the publishing of the article!
Kind Regards,
Your Reviewer
Comments on the Quality of English LanguageThe article is written very well. English is at a very high level. My comment 1 is the only and very minor issue related to the Language.
Thank you!
Round 2
Reviewer 3 Report
Comments and Suggestions for Authors
Dear Authors,
Well done! You did an outstanding job in addressing the comments and further improvement of the article. I recommend it for publishing.
It was a pleasure to review such a great and innovative article. I wish you great success in your future research and careers.
Kind Regards,
Your Reviewer